# GSEA and the coexpression network approach identify novel pathway connections of molecular processes affected in Porto-sinusoidal vascular disease

Aishwarya Iyer[1,2], Cenna Doornbos[3], Chris Evelo[1], Martina Kutmon[2], Friederike Ehrhart[1]*

1 Department of Translational Genomics NUTRIM/MHeNs, Maastricht University, Maastricht, The Netherlands, 2 Maastricht Centre for Systems Biology and Bioinformatics(MaCSBio), Maastricht University, Maastricht, The Netherlands, 3 Medical BioSciences Department, Radboud University, Nijmegen, The Netherlands

* friederike.ehrhart@maastrichtuniversity.nl

## Abstract

### Background

Porto-sinusoidal vascular disease (PSVD) is a complex, rare liver disease characterized by the absence of cirrhosis, with or without the presence of portal hypertension or histological lesions. Given the knowledge gaps in the mechanisms involved in this disease with unknown etiology, we used omics-based approaches to further elucidate the pathways affected by PSVD, facilitating improvements in the prognosis, diagnosis, and treatment options for these patients.

### Methods

We applied gene set enrichment analysis (GSEA) and weighted gene coexpression network analysis (WGCNA) to identify pathways dysregulated in PSVD. Network construction and visualization were performed in Cytoscape to explore interconnectivity among enriched processes. Within key modules, candidate genes were prioritized by ranking approaches and cross-referenced with findings from previous studies.

### Results and conclusion

In this study using both module eigengene correlation network analysis and GSEA, a novel coordinated dysregulation in PSVD was identified characterized by the simultaneous activation of immune and signaling pathways alongside the suppression of metabolic, ribosomal, and mitochondrial programs, highlighting a critical antagonistic interplay between these systems. Alterations in ribosomal proteins, ATP synthase subunits, and serpin family members highlight translational,

**Data availability statement:** All relevant data are within the manuscript and its Supporting information files.

**Funding:** This work is supported by funding from the European Union's Horizon 2020 research and innovation program under the EJP RD COFUND-EJP N° 825575. There was no additional external funding received for this study. The funders had no role in study design, data collection and analysis, decision to publish, or preparation of the manuscript.

**Competing interests:** The authors have declared that no competing interests exist.

bioenergetic, and anticoagulant dysfunction as core mechanisms. Together, these findings define PSVD as a disorder of integrated immune, vascular, and metabolic imbalance.

## 1. Introduction

Porto-sinusoidal vascular disease (PSVD) is a complex, rare liver disease characterized by the absence of cirrhosis, with or without the presence of portal hypertension or histological lesions [1]. This term was recently coined to improve the understanding of the disease by reducing the effect of heterogeneity, facilitating improved diagnosis, and simplifying comparisons between different clinical studies [1]. PSVD is a rare disease with a currently unknown prevalence.

The diagnosis of patients suffering from PSVD includes noninvasive imaging methods focused on splenomegaly and portosystemic collaterals and hepatic vein venography. However, imaging by itself is insufficient, and further invasive methods, such as biopsies, are an essential part of the diagnostic routines for PSVD [1]. The accuracy of analysis remains highly variable depending on the experience of the histopathologist [1]. In a recently published metabolomics study, a group of metabolite markers was identified that could predict patients diagnosed with PSVD with an accuracy of 88% [2].

While the precise etiology of PSVD remains unknown, its development is strongly linked to vascular alterations within the liver and is frequently associated with disorders of immunity, blood diseases, prothrombotic conditions, and drug exposure [1], suggesting a multifactorial origin involving diverse biological processes. 43–48% of patients with PSVD patients have one or more associated conditions majorly classified into disorders of immunity, blood diseases and prothrombotic conditions, infections, congenital or familial defects, and drug exposure [1].

The small and heterogeneous nature of PSVD patient groups, coupled with varying physiological and histological features, has resulted in sparse information regarding the molecular pathways or processes affected in this condition. A recent study by Hernández-Gea et al., revealed previously unknown regulatory pathways affected in PSVD using co-expression analysis using gene expression data from healthy, PSVD and liver cirrhosis patients. The study indicated deregulation of pathways specific to vascular homeostasis and oxidative phosphorylation affecting the endothelial function [3]. In this study, we aim to understand the connections between the dysregulated pathway/processes in specifically PSVD vs histologically healthy liver tissues using complementary approaches such as GSEA and coexpression network analysis. Building upon these findings, our integrated approach would reveal a more complex, multi-systemic dysregulation, uncovering novel relationships, and refining the understanding of previously implicated pathways like oxidative phosphorylation.

Omics analysis, especially transcriptomics, has been widely used to understand genes differentially regulated in a disease and subsequently linking these genes to pathways to explain the molecular mechanisms underlying the disease. Also, other approaches based on network algorithms, especially co-expression networks, have

been constructed from omics data to identify novel disease-specific mechanisms by identifying genes that are coexpressed or exhibit coordinated changes [4,5].

In this study, we implemented two methods: first, gene set enrichment analysis, and second, co-expression network analysis using transcriptomics to identify pathways or processes affected in patients with PSVD. Understanding the pathways or processes would shed light on the mode of action of the disease, thereby allowing for improved prognosis, diagnosis, and the treatment options available to the patients suffering from this rare disorder.

## 2. Materials and methods

### 2.1 Data

A previously published transcriptomics dataset by Lozano et al. was obtained from the GEO database (GEO:GSE77627) [3]. The dataset contains liver mRNA expression profiles for histologically normal liver (HNL), PSVD and liver cirrhosis patients. In this study, liver cirrhosis patients were excluded given that their transcriptomic profile overlapped with the PSVD transcriptomic profile (see S1 Fig). Additional clinical data and information were obtained from the original study authors, Hospital Clinic of the University of Barcelona. The measured variables included information on sex, wedged hepatic vein pressure (WHVP), hepatic venous pressure gradient (HVPG), bilirubin, platelet count, spleen size, liver stiffness, PSVD-specific, and non-specific biopsy markers.

### 2.2 Data pre-processing

The raw Illumina probe data using the Illumina HumanHT-12 DASL 4.0 R2 expression beadchip platform annotation was first filtered for protein-coding genes using the biomaRt(v2.64.0) R package with the filters: biotype (protein coding), chromosome name (22 chromosomes, mitochondrial chromosome and sex chromosomes) using the gene identifiers provided in the annotation file. Next, the filtered probe data was pre-processed using the lumi(v) R package [6]. Background correction and quantile normalisation was performed using the *neqc* function with an offset value of 16. The data was re-annotated using ENSEMBL gene identifiers. A misdiagnosed patient (PSVD17) was removed from the analysis. Samples with incomplete sex information were removed from the analysis. The data distribution for before and after normalized data is provided in S2 Fig.

To detect outliers, hierarchical clustering on the samples was performed and the dendrogram is provided in S3 Fig.(a). Based on the clustering, sample 'PSVD05' was removed from further analysis. The dendrogram of samples and clinical variables measured for these samples is provided in S3 Fig.(b).

### 2.3 Differentially expressed gene (DEG) analysis

Differential gene expression analysis was performed to determine genes that are significantly altered (up- or down-regulated) in PSVD patients compared to healthy controls after sex correction using the limma(v3.64.1) R package [7]. The cut-off for significantly up-regulated genes is logFC > 0.58 and adjusted p-value < 0.05. For down-regulated genes, the cut-off used is logFC < –0.58 and adjusted p-value < 0.05. The *EnhancedVolcano* function in EnhancedVolcano(v1.26.0) R package was used for creating the volcano plot for differentially regulated genes identified in PSVD vs HNL comparison [8].

### 2.4 Gene set enrichment analysis for DEGs

Gene set enrichment analysis (GSEA) was performed using the clusterProfiler(v4.6.2) R package using a maximum and minimum gene set sizes of 500 and 10 respectively [9–11]. The DEGs were ranked based on the product of signed log fold change and the negative logarithm of the adjusted p-value, see the below.

$$ranking = log_2FC * -log_{10}adjusted\,p-value \tag{1}$$

For the enrichment analysis, the human canonical pathway gene sets from the Molecular Signatures Database (MSigDB, v2023.2.Hs) were used ([10,12]). The pathway genesets from Kyoto Encyclopedia of Genes and Genomes (KEGG, 186 gene sets ([13,14])), WikiPathways (733 gene sets ([15])) and Reactome (1,654 gene sets ([16])) were included. Additionally, 7,751 gene sets from the Biological Process ontology from Gene Ontology (GO) were included for the analysis ([17,18]).

## 2.5 Cytoscape visualisation of the enrichment analysis

The gene set enrichment analysis results were visualized in Cytoscape using a custom R script. To calculate similarity between two enriched terms the overlap coefficient (k) was used. [19].

$$k = \frac{|A \cap B|}{|min(A, B)|}$$

(2)

A cut-off score of $k > 0.4$ was used to add an edge between two enriched terms.

## 2.6 Coexpression network construction

The weighted gene co-expression analysis (WGCNA) is a network algorithm tool that constructs correlation networks based on similar gene expression patterns across samples. It uses an unsupervised approach to identify co-expression gene modules. This tool was implemented using the WGCNA(v1.73) R package to identify gene expression modules correlating to the PSVD phenotype [20].

Normalized gene expression data was adjusted for sex effects using the *removeBatchEffect* function from the limma(v) R package [7]. Next, lowly expressed genes, i.e., genes with average expression values below 0.05 were removed. The input was the pre-processed normalized data of all the samples used (healthy and PSVD). A step-by-step method was used to generate the consensus network and to further detect the modules.

Firstly, a similarity network was constructed using Pearson correlation for all gene pairs in the dataset. Next, a signed adjacency matrix was calculated by raising the similarity matrix to a soft-thresholding power ($\beta = 18$).

Next, the adjacency matrix was converted into a Topological Overlap Matrix (TOM). The TOM is a robust network similarity measure by calculating the effect of neighboring nodes on pairs of genes. The resulting proximity matrix is then converted to a dissimilarity TOM matrix. The dissimilarity measure works well in the clustering of gene expression profiles by identifying distinct gene modules.

From these results, a dendrogram was constructed using the dissimilarity matrix and average hierarchical clustering method (see S4 Fig). To identify modules with highly interconnected genes, a dynamic tree-cut method was implemented with a minimum module size of 150.

## 2.7 Identification of key clinically significant modules and core genes for the key modules

The consensus co-expression network generated previously was then used to identify modules of highly interconnected genes or genes with a higher degree of co-expression using the dynamic tree-cut algorithm. To identify key modules relevant to clinical phenotypes associated with PVSD their corresponding module eigengene and the module membership were calculated. Finally, a significant correlation between the modules and the clinical phenotype 'Diagnosis of PSVD' was used to identify key gene modules. Core genes from significant modules were retained based on thresholds of gene significance $> 0.5$ and $\left| \text{module membership} \right| > 0.5$.

## 2.8 Over-representation analysis (ORA) and functional annotation for key modules

Functional analysis of the core genes for the key clinically significant modules was performed using the *enricher* function in the clusterProfiler R package [9]. The Gene Ontology: Biological Process geneset for performing the

over-representation analysis was obtained from the Molecular Signatures Database (MSigDB, v2023.2.Hs)([10,12,17,18]). For certain modules where enriched terms were not obtained using the above gene set, WikiPathways and the Reactome gene sets were used for performing enrichment analysis. The key modules were then manually functionally annotated by assigning an appropriate biological term based on the enriched terms for each module.

### 2.9  Module Eigengene Correlation Network

Pearson correlation using the *cor* function from the stats(v3.6.2) R package was implemented [21]. The Pearson correlation of the module eigengenes obtained from coexpression network analysis was calculated for the key modules. The network was then exported to Cytoscape and the key modules (as nodes in the network) were annotated using the results from Section 2.8.

### 2.10  Open source code

All analysis steps described were fully automated, and the scripts used for this study are available on WorkflowHub for reproduction and further exploration: https://workflowhub.eu/workflows/1040?version=1.

## 3.  Results

### 3.1  Differentially expressed genes in PSVD

The raw transcriptomics data consisting of 26,776 Illumina probes was processed to correspond to 15,551 protein-coding genes (annotated with ENSEMBL gene identifiers). DEG analysis identified 3,955 significantly up-regulated genes and 3,200 significantly down-regulated genes in PSVD patients.

   The top 3 up-regulated genes were erythropoietin (EPO) - logFC 4.8, Ankyrin Repeat Domain 1 (ANKRD1) - logFC 4.8 and G Antigen 12J (GAGE12J) - logFC 4.5 while the top 3 down-regulated genes were ephrin A2 (EFNA 2) logFC −5.1, Nuclear Factor I C (NFIC)- logFC −5.0 and meteorin, glial cell differentiation regulator (METRN) - logFC −4.9 (Fig 1).

### 3.2  Gene set enrichment analysis

Gene set enrichment analysis (GSEA) was performed on the differentially expressed genes in PSVD. Using the Gene Ontology, KEGG, Reactome and WikiPathways gene sets, 69 significantly enriched terms were obtained, in which 9 positively enriched terms and 60 negatively enriched terms were visualised in Cytoscape (Fig 2; complete results in S1 Table). The top positively significantly enriched term was the Olfactory signaling pathway (Reactome) with a normalised enrichment score (NES) of 2 while the Peroxisome pathway (KEGG) was the top negatively significantly enriched term with a NES of −1.9. A similarity threshold of 0.4 was applied to cluster the enriched terms based on the number of genes common between the terms, thereby sharing similar functional profiles.

   From Fig 2 we observe that the lipid and fatty acid metabolism processes, insulin-related processes, tissue and epithelial cell migration processes, endoplasmic reticulum and Golgi associated processes, organic compounds and tricarboxylic acid metabolic process, organic hydroxy compound metabolic process, monosaccharide metabolic process, glycerolipid metabolic process, nucleoside biphosphate metabolic process, immune-related processes like neutrophil degranulation, viral process, SARS-CoV infection, and protein translation were negatively enriched. Processes related to olfactory and sensory stimulus, sleep regulation, meiosis cell cycle and retinoblastoma were positively enriched.

### 3.3  Identification of 15 key PSVD modules using co-expression network analysis

The co-expression network was constructed using 15,551 protein-coding genes from 27 liver biopsy samples (11 healthy and 16 PSVD patients) using the 'WGCNA' R package. The patients with PSVD included in this study have clinical signs of portal hypertension. The two most frequent signs of portal hypertension in PSVD patients being splenomegaly and the

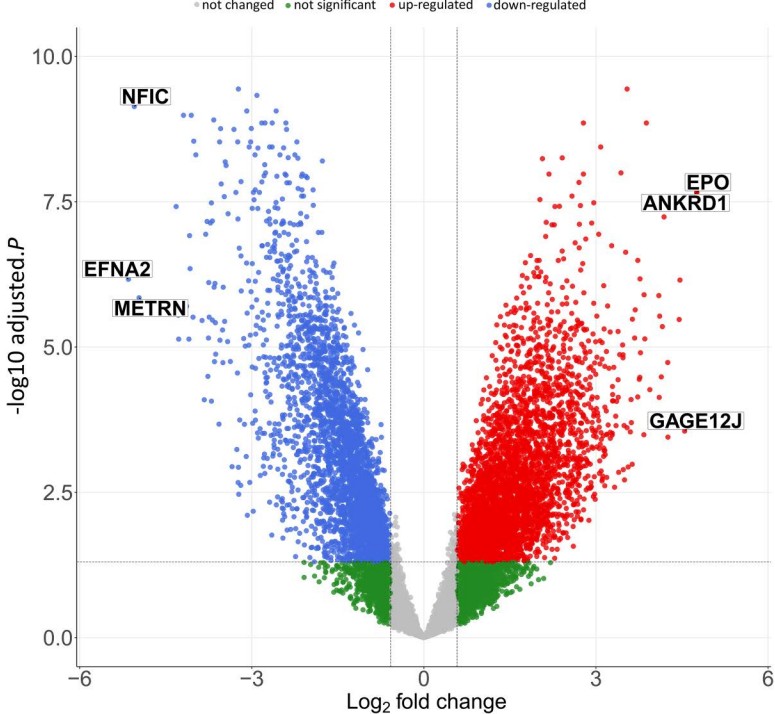

**Fig 1. Volcano plot for differential gene expression.** Volcano plot of all the differentially expressed genes between PSVD and healthy samples in liver tissues. Dots represent individual genes, and the color represents the non-significant genes – grey, non-significant genes with −0.58 < logFC > 0.58 and adjusted p- value > 0.05 – green, significant up-regulated genes with logFC => 0.58 and adjusted p-value < 0.05 – red, significant down-regulated genes with logFC <= −0.58 and adjusted p-value < 0.05 – blue. The top six differentially regulated genes are highlighted in the figure.

presence of gastroesophageal varices. Splenomegaly was present in all PSVD patients with a mean size of 15.3 ± 2.7 cm (Table 1). Additionally, 68% of PSVD patients show clinically elevated portal hypertension with a mean hepatic venous pressure gradient (HVPG) of 7.9 ± 3.8 (Table 1).

A range of soft-thresholding powers ($\beta$) were used to assess the scale free topology of the network constructed. For this analysis, $\beta$ of 18 was selected which had a scale-free topology fit ($R^2$) of 0.81; shown in Fig 3(a). Using the average hierarchical clustering and dynamic tree cut method, a total of 22 distinct gene modules and the corresponding coex-pressed genes for each module were identified. To explore the relationship between the identified coexpressed modules and the clinical variables associated with PSVD phenotype, such as diagnosis, sex, gastroesophageal varices, spleen size, HVPG, WHVP, platelet count, PHT-specific, and PSVD-specific histological markers. Out of the 22 distinctly identified modules, 15 modules were selected which significantly correlated to the diagnosis of PSVD, shown in Fig 3(b) see S5 Fig. for module-trait relationship for all 22 modules).

### 3.4 Selection of core genes in PSVD-associated modules

Core genes for each module were selected based on a threshold of 0.5 for both Gene Significance (GS) and Module Membership (MM). Scatter plots of GS versus MM for each module are shown in S6 Fig. The lightyellow module was excluded from downstream analyses primarily because none of the genes with the module met the 0.5 cut-off for GS, and additionally, the module showed a low correlation with PSVD diagnosis ($r = −0.40$). Fig 4 illustrates the distribution of genes and core genes across modules, as well as their expression profiles.

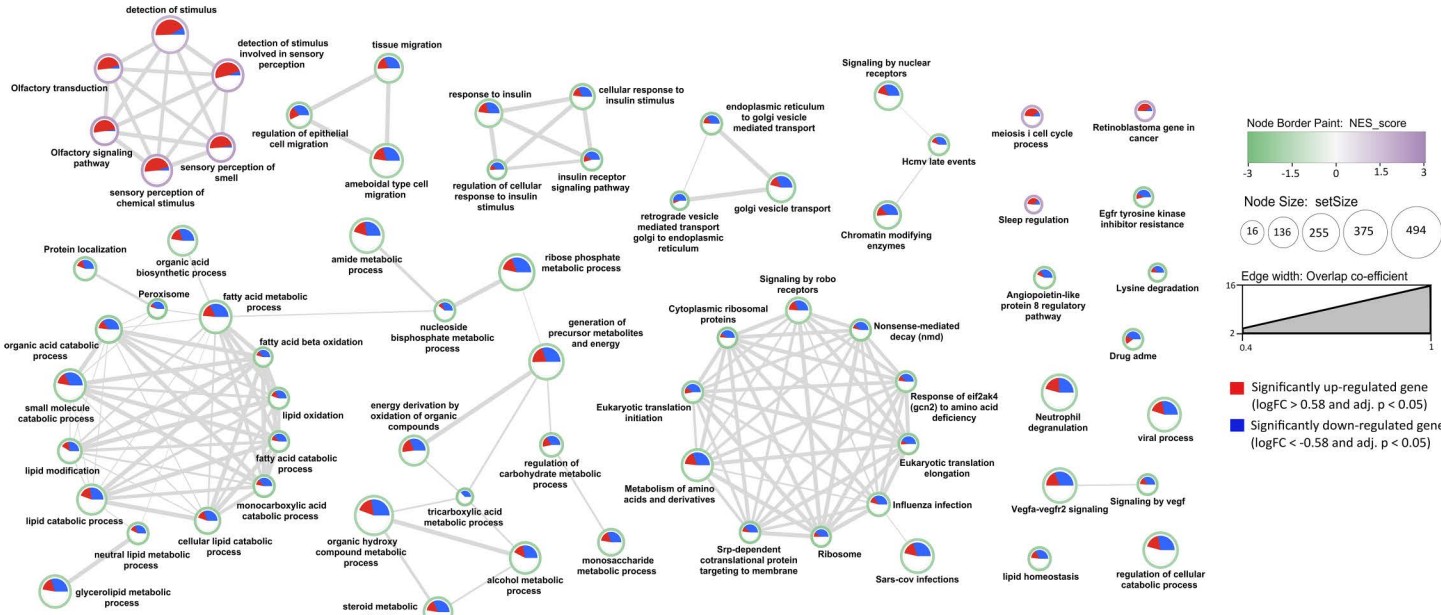

**Fig 2. Gene set enrichment analysis (GSEA) results.** A node represents each enriched gene set of the Gene Ontology class Biological Process and canonical pathways with a false discovery rate cut-off of < 0.05. The node border color indicates normalized enrichment scores of the terms. The pie chart displayed within the node indicates the number of significantly up-regulated and down-regulated (red and blue, respectively) out of the total genes in a gene set. The node size is assigned based on the setSize (number of genes in a term). The edge weight representing the overlap coefficient (similarity index between two terms) has filtered with a cut-off value of 0.4.

**Table 1. Clinical characteristics.**

|  | HNL (n = 11) | PSVD (n = 16) |
|---|---|---|
| Sex(Male) | 4(36%) | 11(69%) |
| Presence GEV | 0 | 5(31%) |
| HVPG (mmHg) | 3.9 ± 0.8 | 7.9 ± 3.8 |
| WHVP (mmHg) | 7.0 ± 0.7 | 13.3 ± 4.1 |
| Platelet Count ($10^9$/l) | 236.5 ± 63.8 | 163.6 ± 143.0 |
| Total Bilirubin (mg/dl) | 0.8 ± 0.4 | 1.2 ± 1.2 |
| Spleen size (cm) | 9.5 ± 0.7 | 15.3 ± 2.7 |
| Liver stiffness (kPa) | 5.8 ± 0.9 | 7.8 ± 3.1 |
| Direct Bilirubin (mg/dl) | 0.4 ± 0.2 | 0.4 ± 0.3 |

[1]Mean ± SD.

[2]GEV, Gastroesophageal varices; HVPG, hepatic venous pressure gradient; WHVP, wedged hepatic venous pressure, HNL, healthy normal liver; PSVD, porto-sinusoidal vascular disease.

## 3.5 Module Eigengene Correlation Network analysis identifies immune, signaling, and metabolic pathways as core dysregulated modules in PSVD

To understand the biological processes affected in PSVD, we performed functional over-representation analysis for the 14 selected modules using the Gene Ontology – Biological Process gene sets from MSigDB. grey60 module with the highest positive correlation and significance ($r = 0.91, p = 6.255351e - 11$) to PSVD diagnosis trait identified processes related to

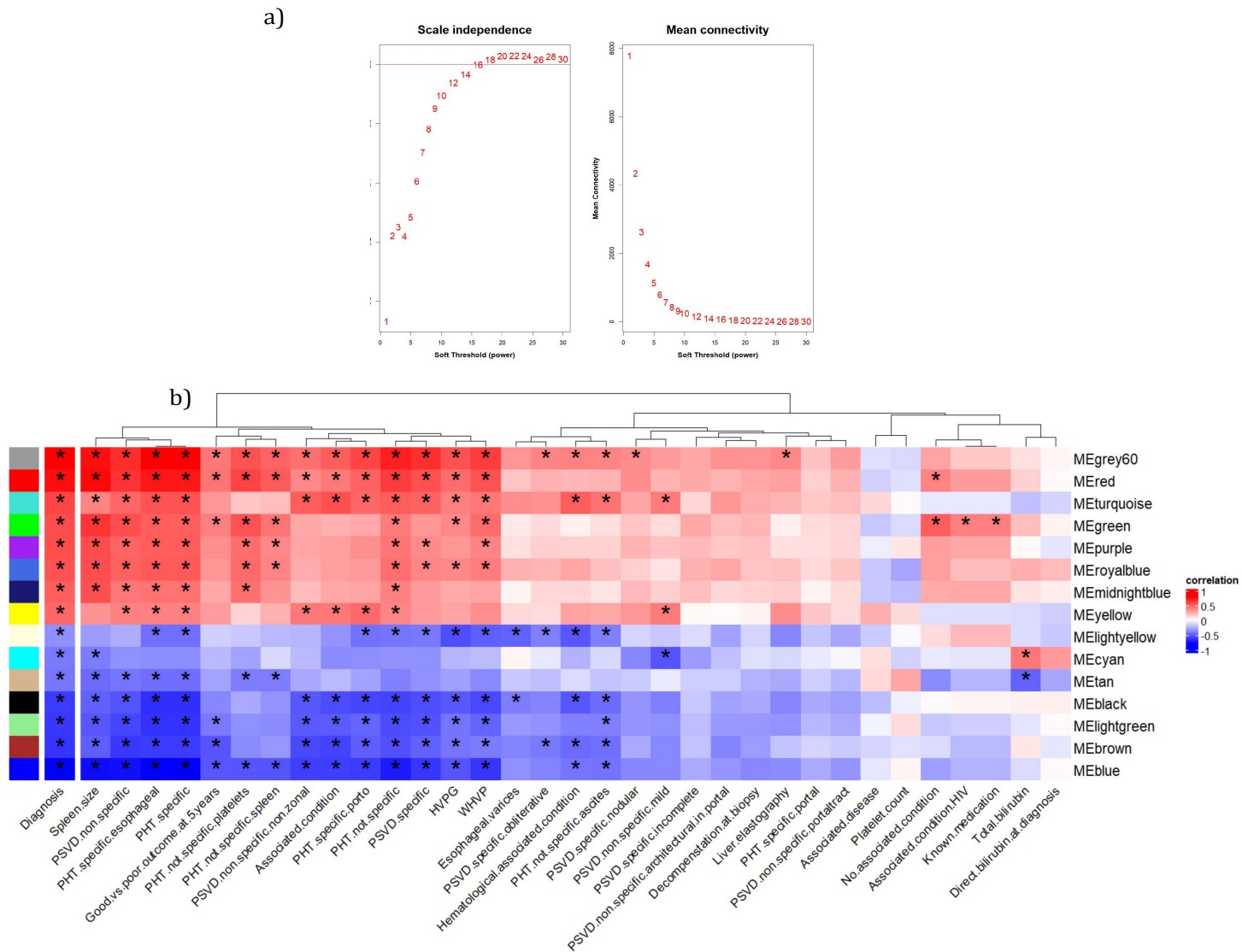

**Fig 3. Scale-free topology and module trait relationship for the coexpression network. (a)** Determination of soft thresholding power for coexpression network construction: 1) Analysis of scale-free index for a range of soft thresholding values ($\beta$). 2) Analysis of the mean connectivity for a range of soft thresholding values ($\beta$). **(b)** A heatmap of the module-trait relationship for key modules significantly correlating to the diagnosis of PSVD on the y-axis and clinical variables on the x-axis. The color gradient on the heatmap represents the strength of the Pearson correlation coefficients. * represents the modules significantly correlating to the respective clinical variables (p < 0.05). Hepatic venous pressure gradient (HVPG) wedged hepatic vein pressure (WHVP), portal hypertension (PHT), porto-sinusoidal vascular diseases (PSVD).

immune cell activation and differentiation, involving the T cells, leukocytes and lymphocytes. This module highlights the adaptive immune system related processes due to terms involving acute and antigenic inflammatory response, positive regulation of cell-cell adhesion and cytokine signaling pathways.

Blue module with highest significant negative correlation ($r = -0.91, p = 6.298297e - 11$) to PSVD diagnosis is strongly enriched for metabolic, translational and insulin-responsive processes. This module reflects the core metabolic functions associated with liver tissue. Key metabolic processes part of the blue module enrichment are fatty acid $\beta$-oxidation, lipid catabolism and biosynthesis, steroid and ketone metabolism, and cellular energy production via oxidative phosphorylation

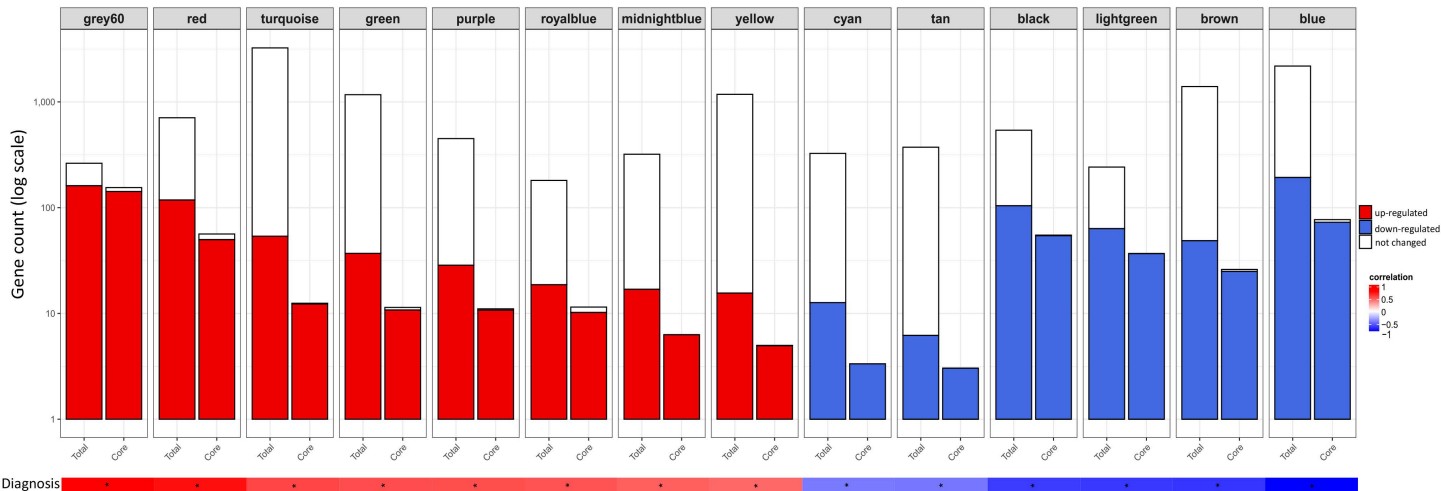

**Fig 4. Distribution of differentially expressed genes within key modules associated with PSVD diagnosis.** Stacked bar plot showing the composition of total and core genes across 14 key modules significantly correlated with PSVD diagnosis. For each module, bars represent total genes (Total; all genes within the module) and core genes (Core; genes filtered by gene significance and module membership threshold > 0.5). Gene counts (y-axis, log10 scale) are categorized by differential expression categories: up-regulated (red), down-regulated (blue), and unchanged (white). The diagnosis correlation bar below indicates positive (red) or negative (blue) association of each module with PSVD diagnosis based on module-trait relationship analysis.

and aerobic respiration processes. Additionally, enriched terms related to cytoplasmic translation, protein ubiquitination and TOR signaling regulation suggest the nutrient sensing, energy balance, and stress response role of liver. Vesicle transport and ER-Golgi trafficking terms hint at active protein and lipid processing, critical in hepatocytes.

Fig 5 presents the Module Eigengene Correlation Network for key modules significantly associated with PSVD diagnosis. The immune-related modules, grey60 (Immune cell Activation and Adhesion) and red (Innate Immune & Vitamin Biosynthesis), exhibit a strong positive correlation with each other, indicating coordinated expression patterns, and both modules show a positive association with PSVD diagnosis. Metabolic modules, including blue (Lipid Metabolism, Energy Production & Insulin Signaling), black (Hepatic Detoxification, and Amino Acid & Lipid Catabolism), and lightgreen (Glycoprotein Metabolism & Endothelial Regulation), also display strong positive correlations among themselves; however, they are negatively associated with PSVD, suggesting a potential downregulation of metabolic pathways in PSVD patients. Signaling-related modules, turquoise (Chemosensory & ciliary motility) and yellow (Sensory Perception & GPCR Signaling), show a strong positive correlation with each other and are positively associated with PSVD, indicating co-regulated signaling processes in PSVD patients. Interestingly, there is a pronounced negative correlation between the metabolic modules (black, blue, lightgreen) and the immune modules (grey60, red), as well as between the metabolic modules and the signaling modules (turquoise, yellow). These patterns reveal the existence of three functionally distinct module clusters: immune, metabolic, and signaling clusters with opposing correlations to the diagnosis of PSVD. This organization may reflect antagonistic or complementary regulatory mechanisms, suggesting that upregulation of immune and signaling pathways may occur concurrently with downregulation of metabolic pathways in PSVD pathogenesis.

### 3.6 Protein-Protein Interaction (PPI) networks for the Immune, Signaling and Metabolic module clusters

Core genes from the immune (grey60 and red) and signaling (yellow and turquoise) module clusters identified in section 3.3 were exported to Cytoscape using the Ensembl identifiers with a STRING confidence score of 0.7 [22]. For the metabolic module cluster, the core genes from black, blue, and lightgreen modules were exported to Cytoscape using Ensembl

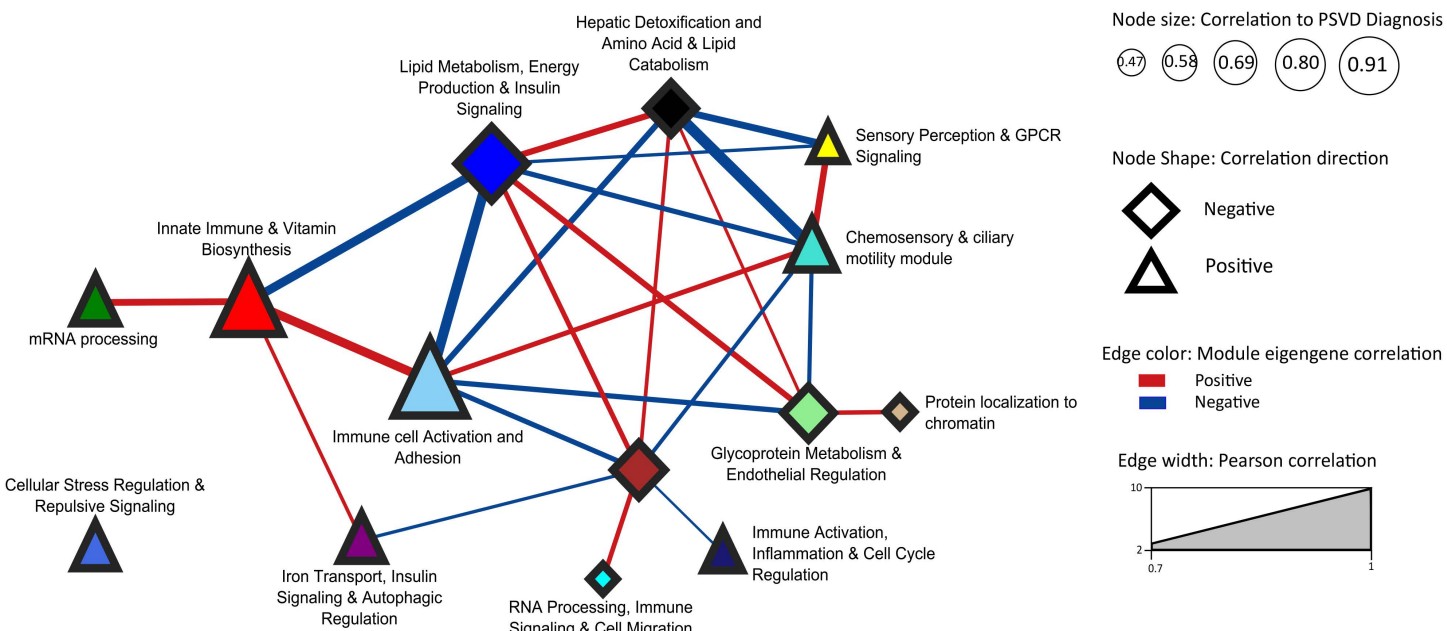

**Fig 5. Module Eigengene Correlation Network of PSVD-associated key modules.** The network visualization depicts the correlations among module eigengenes for modules significantly associated with PSVD diagnosis, highlighting pathways relevant to PSVD. Node size reflects the strength of correlation with PSVD (larger nodes indicate higher correlation, ranging from 0.47 to 0.91), while node shape indicates the direction of the correlation (triangle = positive, diamond = negative). Edges represent correlations between module eigengenes, with color indicating the correlation direction (red = positive, blue = negative) and edge width proportional to the Pearson correlation strength (0.7–1.0). Modules are annotated with enriched biological functions based on Gene Ontology: Biological Processes, providing insights into the functional relevance of PSVD-associated co-expression modules.

identifiers with a STRING confidence score of 0.9 [22]. The confidence threshold was set to 0.7 (high confidence), respectively 0.9 (very high confidence) if the network size becomes too large. Next, Glay community detection algorithm using the Cluster Network option in the clusterMaker app in Cytoscape was implemented to detect clusters within the PPI networks [23,24]. Clusters with less than 20 nodes were removed from the network. The top five genes in the network were selected based on the ranking measure of the product of the absolute log fold change and the node degree.

Fig 6a, representing the PPI network of the immune module cluster for grey60 and red module core genes, indicates five sub clusters identified by the Glay community detection algorithm and functionally annotated using the STRINGapp enrichment function.The large red-bordered node, A2M, has been previously linked to PSVD pathogenesis in the study by Hernández-Gea et al. [3]. The large black-bordered nodes (CCR7, JUN,BARD1, RPS27A, and SNRPG) are the top five ranked genes based on the product of node degree and gene log-fold change.

Fig 6b, representing the PPI network of the signaling module cluster for turquoise and yellow module core genes, indicates six sub clusters identified by the Glay community detection algorithm and functionally annotated using the STRINGapp enrichment function. The large red-bordered node, SERPINA12, has been previously linked to PSVD pathogenesis in the study by Hernández-Gea et al. [3]. The large black-bordered nodes (GHRL, TOP2A, CDC6, IL6, and CD19) are the top five ranked genes based on the product of node degree and gene log-fold change.

S7 Fig represents the PPI network of the metabolic module cluster (blue, black, and lightgreen). The network shows 11 sub clusters identified by the Glay community detection algorithm and functionally annotated using the STRINGapp enrichment function. The large red-bordered node, (ATP5MG, ATP5PF, ATPV0C, ATP5F1C, ATP6V0E1, ATP5PO, ATP5F1A, SERPIND1, APOE and APOA2), has been previously linked to PSVD pathogenesis in the study by

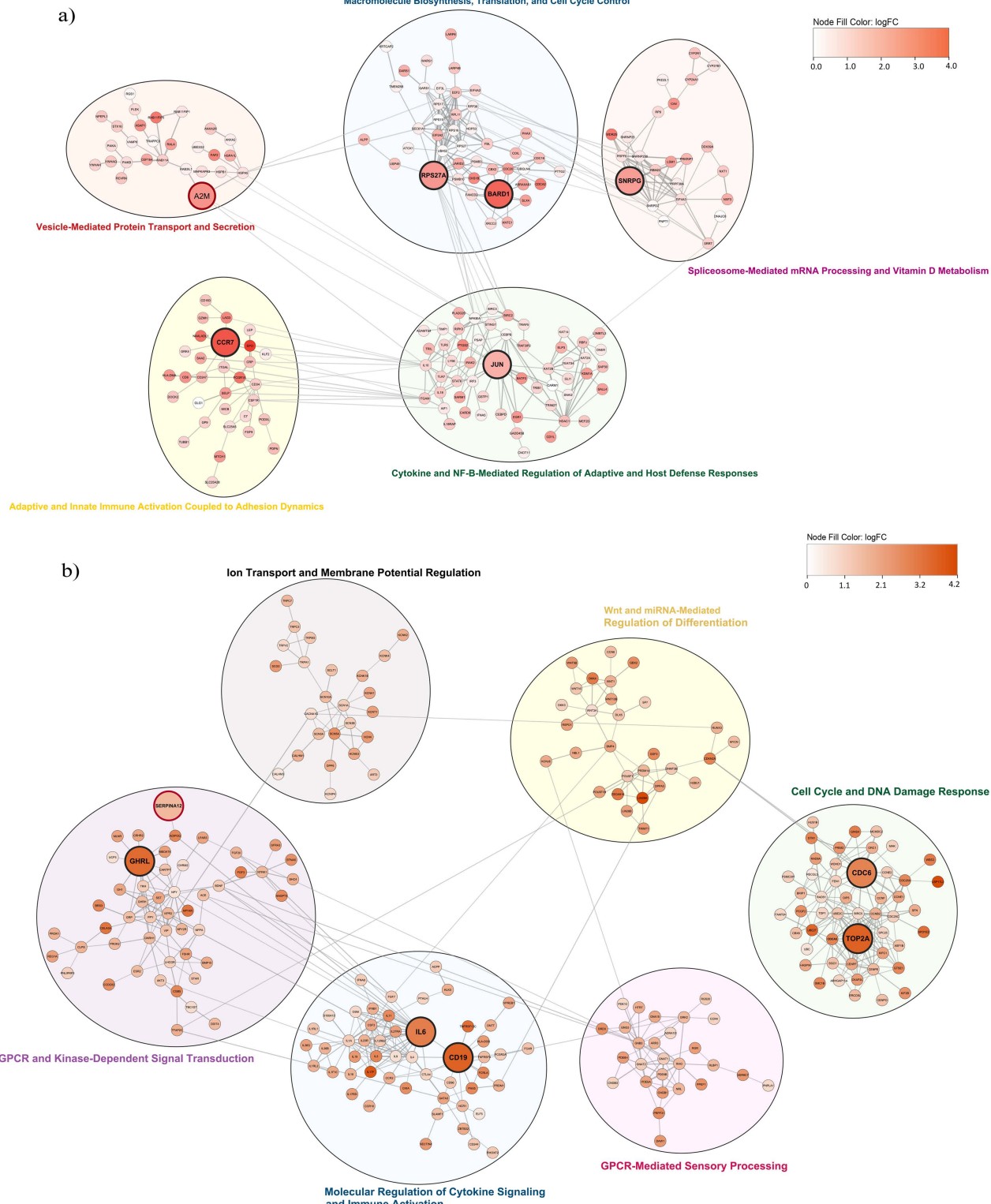

**Fig 6. Protein–Protein Interaction (PPI) networks for immune and signaling module clusters. a)** Immune module cluster (grey60 and red): Nodes represent core genes, colored by log-fold change (logFC). The red-bordered node is a previously reported gene, while large black-bordered nodes are the top five genes ranked by absolute(logFC) × node degree. Subclusters (circles) were identified using the Glay community detection algorithm, with functional enrichment and annotation performed via STRINGapp. **b)** Signaling module cluster (turquoise and yellow): Nodes represent core genes,

colored by logFC. The red-bordered node is a previously reported gene, while large black-bordered nodes are the top five ranked genes. Sub clusters and functional enrichment and annotation were done as above.

Hernández-Gea et al. [3]. The large black-bordered nodes (RPS6, RPS8, RPS11, MRPL12 and RPL12) are the top five ranked genes based on the product of node degree and gene log-fold change.

## 4. Discussion

In this study, we re-analyzed a transcriptomics dataset on PSVD originally produced by Lonzano et al. and focused on the in-depth comparison of transcriptomic changes between the PSVD and HNL groups.

### 4.1 Top differentially expressed genes indicate early cellular stress and vascular dysfunction in PSVD

Upregulated EPO, ANKRD1, and GAGE12J suggests the activation of erythropoietic and mechanotransductive stress pathways [25–31]. Elevated EPO likely reflects IL-6–dependent hepatic signaling under hypoxic or inflammatory conditions [32]. ANKRD1, a YAP/Hippo-responsive mechanosensor induced by pro-inflammatory cytokines, may mirror endothelial strain and extracellular-matrix remodeling, both central to sinusoidal injury [33]. GAGE12J, lacking functional annotation, of potential relevance to PSVD warrants for future experimental validation to elucidate its role in disease pathogenesis.

Downregulated EFNA2, NFIC, and METRN indicate impaired vascular stability and regeneration. EFNA2 loss may weaken angiogenic and immune–endothelial communication [34–36]. Unlike hepatocellular carcinoma, where EFNA2 is upregulated and pro-angiogenic [37], this downregulation may represent a PSVD-specific maladaptive response of the portal microcirculation. NFIC reduction suggests diminished hepatocyte proliferation and matrix regulation through TGF-$\beta$-dependent signaling [38,39]. METRN downregulation implies disturbed endothelial–immune signaling and vascular repair [40]. Together, these genes define early molecular events linking inflammation, vascular stress, and regenerative failure.

### 4.2 GSEA and WGCNA reveal dysregulation in immune and metabolic clusters as potential mechanistic cause

Using both GSEA and WGCNA, we examined altered processes in PSVD and their interrelationships to gain mechanistic insights into disease pathogenesis. The Module Eigengene Correlation Network (see Fig 5) highlighted three major clusters. The immune cluster (immune cell activation, adhesion (grey60) and innate immune and vitamin biosynthesis (red) modules) was positively associated with PSVD. The signaling cluster (chemosensory and ciliary motility (turquoise) and sensory perception and GPCR signaling (yellow) modules) was also positively associated with PSVD, consistent with GSEA findings. In contrast, the metabolic cluster (hepatic detoxification, amino acid and lipid catabolism (black), lipid metabolism, energy production, and insulin signaling (blue), and glycoprotein metabolism with endothelial regulation (lightgreen) modules) was negatively associated, corroborated by GSEA. Importantly, immune and signaling modules were positively correlated with each other but negatively correlated with metabolic modules. This pattern suggests a coordinated dysregulation in PSVD, where heightened immune and signaling activity occurs in parallel with the suppression of metabolic pathways, highlighting novel pathway interconnections that may underlie disease mechanisms.

### 4.3 Methodological and biological insights from GSEA versus WGCNA comparison

Interestingly, immune enrichment was not detected by GSEA, despite being clearly identified through WGCNA co-expression analysis. This discrepancy reflects fundamental differences in analytical approaches and provides complementary biological insights. GSEA operates by testing predefined gene sets for coordinated directional changes (up- or down-regulation) and is optimized to detect strong, uniform shifts in pathway activity. In contrast, WGCNA identifies modules based on co-expression patterns without requiring consistent directionality, capturing subtle or bidirectional immune

dysregulation that may not meet GSEA statistical thresholds. The identification of immune modules by WGCNA suggests that PSVD involves complex, potentially heterogeneous immune responses where individual genes within immune pathways may exhibit varied expression changes. Biologically, this finding indicates that immune dysregulation in PSVD may be more nuanced than a simple pro-inflammatory state, involving coordinated co-expression of immune genes that collectively contribute to disease pathogenesis even when individual pathway enrichment is modest. The complementarity of GSEA and WGCNA thus enabled us to capture both strong directional metabolic suppression and subtle, coordinated immune network alterations, providing a more comprehensive understanding of PSVD molecular mechanisms.

### 4.4  Immune module cluster network

Within the immune module cluster network (see Fig 6a), we identified the top five ranked genes based on their node degree and log fold change. No direct link between BARD1 and PSVD has been reported; BARD1 is mainly studied as a BRCA1 partner in DNA repair and is overexpressed in hepatocellular carcinoma, where it promotes tumor progression [41], suggesting it may represent a novel candidate for future validation in PSVD. Although RPS27A has not been linked to PSVD to date, its roles in translational control, ubiquitin signaling, and inflammation via NF-$\kappa$B suggest it is a plausible candidate for influencing endothelial stress or regenerative pathways in the portal microvasculature. No known direct evidence links JUN (c-Jun) to PSVD. However, in the liver, c-Jun regulates hepatocyte survival, proliferation, and fibrosis in injury contexts [42], positioning it as a candidate gene warranting experimental investigation. Though not studied in PSVD, CCR7's role in modulating hepatic T-cell homeostasis and its immunoregulatory function make it a plausible candidate for involvement in portal microvascular inflammation and immune infiltration in PSVD. In a PSVD co-expression network study, A2M was identified as a highly connective gene, suggesting it may contribute to disease pathogenesis through roles in protease inhibition and matrix remodeling [3].

### 4.5  Signaling module cluster network

For the signaling module cluster network (see Fig 6b), the top 5 ranked genes, GHRL, IL6, CD19, CDC6, and TOP2A, have not been directly linked to PSVD. However, their roles in hepatoprotection and fibrosis (GHRL) [43], inflammation and endothelial injury (IL6, CD19) [44], and liver inflammation and immune cell regulation (CDC6, TOP2A) [45] suggest that their dysregulation could contribute to vascular remodeling and impaired regeneration in PSVD, warranting further experimental validation as potential candidate targets. In contrast to Hernández-Gea et al., SERPINC1 was found to be downregulated when comparing PSVD to liver cirrhosis and healthy controls; our PSVD versus healthy liver biopsy analysis revealed SERPINA12 upregulation within the signaling module cluster [3]. Given that serpins act as key regulators of protease activity, coagulation balance, and inflammatory signaling, this pattern suggests that distinct serpin family members may differentially contribute to PSVD pathogenesis. SERPINC1 reflects altered anticoagulant activity, while SERPINA12 may participate in modulating vascular inflammation and endothelial–immune interactions.

### 4.6  Metabolic module cluster network

In the metabolic module cluster (see S7 Fig), the top 5 ranked genes, RPS6, RPS8, and RPS11, part of the translation and ribosome biogenesis subcluster, were downregulated, indicating impaired ribosome biogenesis and translational activity. Ribosomal stress is known to activate p53, disrupt hepatocyte viability, and modulate immune regulation [46–48], suggesting that suppressed ribosomal protein expression may contribute to endothelial dysfunction and microvascular remodeling in PSVD. In PSVD, SERPINC1 (antithrombin III) has been highlighted as a hub gene in co-expression network analysis, underscoring the role of anticoagulant serpins in disease pathogenesis [3]. SERPIND1 (heparin cofactor II), part of the complement, proteolysis, and vesicle-mediated immune regulation subcluster, although not yet studied in PSVD, shares functional overlap with SERPINC1 as a liver-derived serpin that inhibits thrombin activity in the presence of heparin or dermatan sulfate [49,50]. Together, these findings suggest that the dysregulation of multiple anticoagulant serpins may

collectively influence coagulation balance and vascular remodeling in PSVD. APOE and APOA2, part of the complement, proteolysis, and vesicle-mediated immune regulation subcluster in the metabolic module network, were downregulated in PSVD, contrasting with their upregulation in Hernández-Gea et al. [3]. This difference likely reflects variations in the control groups (HNL vs. cirrhosis+HNL), but consistently implicates lipid metabolism in PSVD pathogenesis. In the network, we additionally observed a consistent downregulation of multiple ATPase subunits, including mitochondrial ATP synthase components (ATP5MG, ATP5PF, ATP5F1C, ATP5PO, ATP5F1A) and vacuolar proton pump subunits (ATP6V0E1, ATPV0C), which are part of the mitochondrial energy metabolism and biosynthesis subcluster. This points to deficits in oxidative phosphorylation and vesicular acidification. Notably, Hernández-Gea et al. (2021) also flagged ATP synthases (e.g., ATP5G1, ATP5B) as highly connective genes within the PSVD transcriptomic network, implicating mitochondrial energy dysfunction as a shared pathogenic axis [3]. The coordinated downregulation of ribosomal proteins (RPS6, RPS8, RPS11) and ATPase subunits (ATP5MG, ATP5PF, ATP5F1C, ATP5PO, ATP5F1A, ATP6V0E1, ATPV0C) points to a combined translational and bioenergetic dysfunction. Together with Hernández-Gea et al. (2021) highlighting these pathways as network hubs, this underscores impaired protein synthesis and mitochondrial energy metabolism as central drivers of PSVD pathogenesis.

A limitation of this study is the relatively small patient cohort, a common issue in studying rare diseases. Although pooling transcriptomic data across studies could help, evolving diagnostic criteria for PSVD and INCPH complicate integration. Nonetheless, our re-analysis provides complementary insights to earlier work by Hernández-Gea et al. (2021). By directly comparing PSVD with HNL, we refined existing observations and uncovered additional interconnected processes and candidate genes. A central pattern emerging from our analysis is a coordinated dysregulation in PSVD of the immune and signaling pathways being upregulated in parallel with the suppression of metabolic processes. While these findings describe processes at the population level, patient heterogeneity in risk factors, genetics, and environmental exposures remains a challenge. While the molecular pathways identified in this study, including those derived from GSEA and coexpression network analyses offer insight into the dysregulated biological processes underlying PSVD, directly linking these signatures to established morphologic correlates such as portal venopathy, sinusoidal remodeling, and altered hepatic microcirculation remains challenging in the absence of corresponding histopathologic and hemodynamic data. Whether the identified pathways reflect a primary vascular injury, immune-mediated mechanisms, or secondary remodeling processes is an important question that the current dataset cannot definitively resolve. Notably, recent clinicopathologic observations have begun to characterize PSVD-like alterations in distinct settings, such as transplant-associated portal vasculopathy, highlighting the broader relevance of vascular remodeling signatures across hepatic disease contexts [51]. Future studies integrating transcriptomic findings with detailed histopathologic profiling and clinical vascular phenotyping will be essential to establish how these molecular signatures map onto the clinicopathologic spectrum of PSVD, and to clarify their potential utility in disease diagnosis, prognosis, and the identification of therapeutic targets. Emerging evidence indicates that gut dysbiosis contributes to PSVD onset and progression by driving porto-sinusoidal abnormalities and intrahepatic thrombosis. Disruption of the gut–liver axis permit translocation of microbial products, such as LPS and metabolites, into the liver. Future studies integrating microbiome profiling with transcriptomic and metabolic analyses are needed to uncover gut–liver–vascular interactions in disease pathogenesis [52–54].

## 5. Conclusion

This re-analysis of liver transcriptomics data, focusing on the comparison of PSVD vs. histologically normal liver, uncovers a coordinated imbalance between upregulated immune and signaling and downregulated metabolic, translational, and bioenergetic pathways as the potential main mechanism behind PDVD development. Using both module eigengene correlation network analysis along with the GSEA, we identified a core pathogenic mechanism in PSVD characterized by a coordinated immune-signaling versus metabolic-translational-bioenergetic imbalance. Specifically, heightened immune activation and signaling pathway upregulation occurs in parallel with profound suppression of

core metabolic processes, particularly affecting ribosomal protein expression (RPS6, RPS8, RPS11) and mitochondrial ATP synthase function (ATP5MG, ATP5PF, ATP5F1C, ATP5PO, ATP5F1A, ATP6V0E1, ATPV0C). This combined translational and bioenergetic deficit represents a central pathogenic axis in PSVD, consistent with hub gene identification in prior network analyses. The complementarity of GSEA and WGCNA proved essential given that while GSEA captured strong directional metabolic suppression, WGCNA revealed subtle but coordinated immune network dysregulation not detected by pathway enrichment alone, underscoring the complexity of immune involvement in PSVD. We identified several potential candidate genes, including BARD1, GAGE12J, RPS27A, JUN, and CCR7 in immune and signaling modules, alongside validated genes such as A2M and multiple serpins (SERPINC1, SERPIND1, SERPINA12) involved in anti-coagulation dysnfunction that suggests for future experimental validation to confirm their roles in disease pathogenesis.

Together, these findings position PSVD as a disorder of integrated immune, anti-coagulation, and metabolic dysregulation. They also underscore the need for effective multi-omics studies to validate candidate genes and pathways uncovered in this and previous studies, while emerging evidence on the gut–liver axis suggests that incorporating microbiome analysis may uncover novel methods of non-invasive patient diagnosis and prognosis. This systems-level framework would refine our understanding of PSVD pathogenesis and open up further avenues for mechanistic exploration and biomarker discovery.

## Supporting information

**S1 Fig. Principal component analysis (PCA) for healthy, liver cirrhosis, and PSVD liver biopsy samples.** PCA was performed on normalized transformed gene expression data. The PCA plot above represents the variance explained by the top two components.
(PDF)

**S2 Fig. Distribution of the gene expression data across samples.** (A)Distribution of the gene expression data before and after normalization. X-axis represents the samples (pink – Healthy liver biopsies and blue- PSVD liver biopsies). Y-axis represents the genes expression values in logarithmic scale. (B) Distribution of the detection p-value before and after normalization. X-axis represents the samples (pink – Healthy liver biopsies and blue- PSVD liver biopsies). Y-axis represents the p-values of the probes used to measure the gene expression of the samples.
(PDF)

**S3 Fig. Hierarchical clustering of the samples.** (a) Dendogram of the sample clustering. Sample PSVD05 (shown in red) was removed from the analysis given that it was clustering with the healthy liver samples. (b) Dendogram representing the sample clustering after outlier removal against the clinical variables visualized in the rows.
(JPG)

**S4 Fig. Cluster dendrogram and modules detected.** Hierarchical tree (average linkage) using dynamic tree cutting method was used for module detection. The Dynamic Tree Cut band represents the genes assigned to particular modules. 22 modules were detected.
(PDF)

**S5 Fig. Module-trait relationship heatmap.** A heatmap of the module-trait relationship for modules significantly correlating to the diagnosis of PSVD on the y- axis and clinical variables on the x-axis. The color gradient on the heatmap represents the strength of the Pearson correlation coefficients. Number in each cell is the correlation and the p-value (in brackets). Hepatic venous pressure gradient (HVPG), wedged hepatic vein pressure (WHVP), portal hypertension (PHT), porto-sinusoidal vascular diseases (PSVD).
(PDF)

**S6 Fig. The scatter plots depict the relationship between gene significance (GS) for PSVD diagnosis and module membership (MM) within each key co-expression module identified by WGCNA.** Each point represents a single gene, where GS reflects the correlation between gene expression and PSVD diagnosis, and MM represents the correlation between the gene and the module eigengene, indicating its connectivity within the module. Genes in the upper-right quadrant (GS > 0.5 and MM > 0.5) were designated as core (hub) genes, as they are both strongly associated with the trait and highly central within the module network.Data points are color-coded to reflect differential expression in PSVD: significantly up-regulated (red), down-regulated (blue) and non-significant (grey) genes.
(PDF)

**S7 Fig. Protein–Protein Interaction (PPI) Networks for the Metabolic Module cluster (blue, black, lightgreen).** Nodes represent core genes, colored by log-fold change (logFC). The red-bordered node is a previously reported gene, while large black-bordered nodes are the top five genes ranked by absolute(logFC) × node degree. Subclusters (circles) were identified using the Glay community detection algorithm, with functional enrichment and annotation performed via STRING.app.
(PDF)

**S1 Table Enrichment output for the 14 key modules.**
(CSV)

## Acknowledgments

The authors would also like to thank the data providers Juan Carlos Garcia-Pagan and his team for their support and helpful discussions. Also, we would like to acknowledge Daphne Wijnbergen (ORCID: 0000-0002-7449-6657) for her help in implementing the Common Workflow Language for this study.

## Author contributions

**Conceptualization:** Chris Evelo, Martina Kutmon, Friederike Ehrhart.

**Data curation:** Aishwarya Iyer, Cenna Doornbos.

**Formal analysis:** Aishwarya Iyer, Martina Kutmon, Friederike Ehrhart.

**Funding acquisition:** Chris Evelo, Friederike Ehrhart.

**Methodology:** Aishwarya Iyer.

**Project administration:** Cenna Doornbos.

**Resources:** Cenna Doornbos.

**Supervision:** Chris Evelo, Martina Kutmon, Friederike Ehrhart.

**Writing – original draft:** Aishwarya Iyer.

**Writing – review & editing:** Aishwarya Iyer, Cenna Doornbos, Chris Evelo, Martina Kutmon, Friederike Ehrhart.

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
