## [Decision Letter · Decision Letter 0]

29 Dec 2025

PONE-D-25-56812GSEA and the coexpression network approach identify novel pathway connections of molecular processes affected in Porto-sinusoidal vascular diseasePLOS One

Dear Dr. Ehrhart,

Thank you for submitting your manuscript to PLOS ONE. After careful consideration, we feel that it has merit but does not fully meet PLOS ONE’s publication criteria as it currently stands. Therefore, we invite you to submit a revised version of the manuscript that addresses the points raised during the review process.

We look forward to receiving your revised manuscript.

Kind regards,

Cheung Kenneth Chat Pan

Academic Editor

PLOS One

Journal Requirements:

“This work is supported by funding from the European Union’s Horizon 2020 research and innovation program under the EJP RD COFUND-EJP N° 825575.”

“This work is supported by the funding from the European Union’s Horizon 2020 research and innovation programme under the EJP RD COFUND-EJP N° 825575. The authors would also like to thank the data providers Juan Carlos Garcia-Pagan and his team for their support and helpful discussions.”

“This work is supported by funding from the European Union’s Horizon 2020 research   and innovation program under the EJP RD COFUND-EJP N° 825575.”

6. We notice that your supplementary figures are included in the manuscript file. Please remove them and upload them with the file type 'Supporting Information'. Please ensure that each Supporting Information file has a legend listed in the manuscript after the references list.

7. We note that there is identifying data in the Supporting Information file < Supplementary File 1.csv>. Due to the inclusion of these potentially identifying data, we have removed this file from your file inventory. Prior to sharing human research participant data, authors should consult with an ethics committee to ensure data are shared in accordance with participant consent and all applicable local laws.

-Location data

Additional Editor Comments (if provided):

1. Reconsider the "weak" DEG category. Use standard FDR < 0.05 (or a justified threshold) for calling DEGs. If retaining "weak" DEGs, provide a strong biological or statistical rationale for the chosen logFC cutoffs and clearly differentiate them from significant DEGs in all analyses and visualizations. Avoid including them in functional enrichment summaries unless justified).

2. : For genes without prior direct links to PSVD (e.g., GAGE12J, BARD1, RPS27A, GHRL, IL6, CD19, CDC6, TOP2A), rephrase conclusions. Focus on their statistical ranking and potential biological relevance based on known functions, but avoid labeling them as "novel" findings for PSVD without supporting evidence. Use phrases like "potential candidate", "statistically prioritized", or "suggestive of involvement".

• Ensure all figures are clear, well-labeled, and legends are unambiguous (especially Fig 2 volcano plot colors).

• Critically assess if all panels in Fig 5 are essential. Consider focusing on the core genes (5b).

• Make absolutely certain that Figs 6 and 7 effectively communicate the complex correlation and PPI structures.

3. Elaborate on the GSEA vs. WGCNA immune pathway discrepancy. Discuss the potential biological and methodological reasons.

4. Provide more nuanced biological interpretation of PPI subclusters beyond the functional labels. What specific interactions or potential regulatory roles do the hub genes play within their subclusters in the context of PSVD?

5. If feasible, briefly discuss how specific module-clinical trait correlations (beyond just diagnosis) might inform the understanding of PSVD heterogeneity or severity.

6. Sharpen the conclusion to emphasize the core finding of the immune/signaling vs. metabolic/translational/bioenergetic imbalance as the central pathogenic mechanism revealed by this integrated analysis, alongside the specific ribosomal and mitochondrial deficits. Position the "novel" genes as candidates for future validation.

Reviewers' comments:

Reviewer's Responses to Questions

**Comments to the Author**

1. Is the manuscript technically sound, and do the data support the conclusions?

Reviewer #1: Yes

2. Has the statistical analysis been performed appropriately and rigorously? 

Reviewer #1: Yes

3. Have the authors made all data underlying the findings in their manuscript fully available?

Reviewer #1: Yes

4. Is the manuscript presented in an intelligible fashion and written in standard English?

Reviewer #1: Yes

5. Review Comments to the Author

Reviewer #1: In this work, the authors implemented a secondary analysis of transcriptomics data on Porto-sinusoidal vascular disease (PSVD). Specifically, the authors used both GSEA (gene set enrichment analysis) and WGCNA (weighted gene co-expression network analysis) to identify candidate modules and enriched pathways.

Major Issues:

• I don’t have any major issues with the bioinformatics approach used by the authors. With that being said, the below comments should be addressed to improve the clarity of the manuscript.

• In GSEA, looks like the authors used both significant and weakly significant (p-value > 0.05) genes. What is the rationale behind using the weak significant candidates? Did the authors try using just the significant ones?

• In the WGCNA analysis, it is not clear how many modules the authors were able to find. In [Sec sec022] (Page 4, line 137), the authors specified finding 24 modules. However, in [Sec sec026] (line 227), it’s mentioned that 35 distinct modules were identified. Moreover, in the Supplementary Figure (Module-trait relationship heatmap) I could find only 22 modules. This needs to be clarified.

• For constructing module PPI networks, the authors seem to have used different STRING confidence score thresholds. What is the reason/rationale for this?

• This manuscript can do with another table listing some of the top (relevant) enriched pathways (and individual genes from the DEGs/core genes) from both analyses. This can help identify robust pathways enriched in both methodologies.

Minor comments:

• Looks like the placeholder texts form the journal manuscript template are still present, like Fig 1 caption in Page 6. This needs to be fixed.

• In [Sec sec019], looks like a negative sign is missing in describing the significantly weak downregulated genes. Also looks like the authors missed adding the word “weak” in this line (line 93). This should be corrected.

• Also, for the DEG categories, the authors used “significantly weak upregulated/downregulated” genes as the category names. Perhaps, using the phrase ‘significantly weak’ is confusing since the adjusted p-value thresholds used were not significant (> 0.05).

• The sentence starting in line 228 ([Sec sec026]) seems to be left incomplete. This needs to be fixed.

• Formatting of Figure 5 legend seems to have been messed up in the manuscript.

• In the figure legend for Figure 6, the correlation values seem to have commas instead of decimal points.

• In figure S4 (cluster dendrogram), the label on the x-axis is cut.

• Text on line 9 ([Sec sec005]) seems to have been scrambled. This needs to be fixed.

6. PLOS authors have the option to publish the peer review history of their article (what does this mean?). If published, this will include your full peer review and any attached files.

Reviewer #1: **Yes:** Sudhir Ghandikota

---

## [Author Response · Author response to Decision Letter 1]

19 Feb 2026

Please see the "response to reviewers" file, it contains all responses to Editorial and reviewers comments.

---

## [Decision Letter · Decision Letter 1]

11 Mar 2026

PONE-D-25-56812R1GSEA and the coexpression network approach identify novel pathway connections of molecular processes affected in Porto-sinusoidal vascular diseasePLOS One

Dear Dr. Ehrhart,

Thank you for submitting your manuscript to PLOS ONE. After careful consideration, we feel that it has merit but does not fully meet PLOS ONE’s publication criteria as it currently stands. Therefore, we invite you to submit a revised version of the manuscript that addresses the points raised during the review process.

We look forward to receiving your revised manuscript.

Kind regards,

Cheung Kenneth Chat Pan

Academic Editor

PLOS One

**Journal Requirements:**

**Additional Editor Comments:**

The study presents a comprehensive re-analysis of existing transcriptomics data for Porto-sinusoidal vascular disease (PSVD) using robust bioinformatics methods (GSEA and WGCNA). The findings point to a coordinated dysregulation involving immune, signaling, and metabolic pathways, which is a valuable contribution to understanding this rare disease.

1. Clarity on "Novel Pathway Connections": The title explicitly mentions "novel pathway connections," but the abstract doesn't clearly articulate what these novel connections are. While it mentions "coordinated dysregulation," it could be more explicit about the interrelationships found (e.g., the negative correlation between immune/signaling and metabolic modules).

• In the "Results and Conclusion" section, explicitly state that the study identifies novel interconnections or antagonistic relationships between these pathway clusters. For example: "This study revealed a novel coordinated dysregulation in PSVD, characterized by the simultaneous activation of immune and signaling pathways alongside the suppression of metabolic, ribosomal, and mitochondrial programs, highlighting a critical antagonistic interplay between these systems."

2. "Coordinated dysregulation" is a key finding. Can you hihglgiht at how this coordination was identified (e.g., through module eigengene correlation)? phrase like: "Through module eigengene correlation network analysis, this study revealed a coordinated dysregulation..."

3. While ribosomal proteins, ATP synthase, and serpin family members are mentioned, their significance could be further emphasized. Why are these particular molecules important?explain the consequences of their alteration. For example: "Alterations in ribosomal proteins, ATP synthase subunits, and serpin family members highlight translational, bioenergetic, and anticoagulant dysfunction, respectively, as core mechanisms."

• After introducing Hernández-Gea et al. (lines 23-26), explicitly state the gap your study fills. For example: "While Hernández-Gea et al. provided crucial insights into vascular homeostasis and oxidative phosphorylation, a comprehensive, integrated understanding of the interplay between various dysregulated biological systems in PSVD, particularly the coordinated relationships between immune, metabolic, and signaling pathways, remains elusive. Our study aims to address this by..."

• Clarify the Relationship with Hernández-Gea et al. (Lines 23-26): This is the most direct comparator study. The introduction mentions its findings (vascular homeostasis, oxidative phosphorylation, endothelial function). Briefly mention how your approach differs or goes deeper. For example, did Hernández-Gea et al. use WGCNA but not GSEA, or did they focus on different aspects of network analysis? (The discussion section does address this, but a hint in the intro would be good).

• The discussion (lines 369-372, 394-396, 402-404) highlights differences and similarities with Hernández-Gea et al. (e.g., SERPINs, APOE/APOA2, ATP synthases). Consider weaving a very high-level preview of this into the introduction to further emphasize novelty. For instance, "Building upon these findings, our integrated approach reveals a more complex, multi-systemic dysregulation, uncovering novel antagonistic relationships between immune and metabolic processes, and refining the understanding of previously implicated pathways like oxidative phosphorylation." While true, the introduction could acknowledge the hypothesized contributors to PSVD's etiology, given the mention of associated conditions (lines 17-19). This would subtly connect to the pathways explored.

• Instead of just "The mechanism of disease development for PSVD is not known" (line 15), consider: "While the precise etiology of PSVD remains unknown, its development is strongly linked to vascular alterations within the liver and is frequently associated with disorders of immunity, blood diseases, prothrombotic conditions, and drug exposure [1], suggesting a multifactorial origin involving diverse biological processes." This naturally leads into why omics are needed.

• Line 20: "Because PSVD incorporates a small, heterogeneous diseased patient group with varying physiological and histological features, there is sparse information regarding molecular pathways or processes affected in this condition." This is a slightly long sentence. Consider breaking it or rephrasing for better flow. "The small and heterogeneous nature of PSVD patient groups, coupled with varying physiological and histological features, has resulted in sparse information regarding the molecular pathways or processes affected in this condition."

Ensure consistent use of "Porto-sinusoidal vascular disease" or its abbreviation "PSVD" throughout. The current usage is generally good.

• Line 23: "Hernand´ez-Gea et al." -> "Hernández-Gea et al." (check diacritics).

• Line 28: "and next to link these genes to pathways thereby explaining the molecular mechanisms" -> "and subsequently linking these genes to pathways to explain the molecular mechanisms".

• Line 31: "are coexpressed or change" -> "are coexpressed or exhibit coordinated changes".

Reviewers' comments:

Reviewer's Responses to Questions

**Comments to the Author**

1. If the authors have adequately addressed your comments raised in a previous round of review and you feel that this manuscript is now acceptable for publication, you may indicate that here to bypass the “Comments to the Author” section, enter your conflict of interest statement in the “Confidential to Editor” section, and submit your "Accept" recommendation.

Reviewer #2: (No Response)

2. Is the manuscript technically sound, and do the data support the conclusions?

Reviewer #2: Yes

3. Has the statistical analysis been performed appropriately and rigorously? 

Reviewer #2: Yes

4. Have the authors made all data underlying the findings in their manuscript fully available?

Reviewer #2: Yes

5. Is the manuscript presented in an intelligible fashion and written in standard English?

Reviewer #2: Yes

6. Review Comments to the Author

**Reviewer #2:** This is a well-designed and technically sound study; however, the Discussion would benefit from a focused revision to better contextualize the identified pathways within the evolving clinicopathologic understanding of porto-sinusoidal vascular disease (PSVD). In particular, the authors should more clearly link the GSEA- and coexpression network–derived pathways to established morphologic and clinical correlates of PSVD, including portal venopathy, sinusoidal remodeling, and altered hepatic microcirculation. Briefly addressing how these molecular signatures may overlap with or diverge from PSVD-like changes observed in other settings (e.g., post-transplant portal vasculopathy) would strengthen the translational relevance. The Discussion should also clarify whether the identified pathways support a primary vascular injury model, immune-mediated mechanisms, or secondary remodeling processes. Incorporation of recent clinicopathologic observations would enhance the manuscript, including: Khandakar B, Joldoshova A, Zhang X. Porto-sinusoidal vascular disease-like alterations in transplant liver biopsies: A clinicopathologic observation of transplant-associated portal vasculopathy. Hum Pathol. 2026;167:105984. doi:10.1016/j.humpath.2025.105984.

7. PLOS authors have the option to publish the peer review history of their article (what does this mean?). If published, this will include your full peer review and any attached files.

Reviewer #2: No

---

## [Author Response · Author response to Decision Letter 2]

27 Mar 2026

Please see the response to reviewers in the attached document.

---

## [Editor Report · Decision Letter 2]

1 Apr 2026

GSEA and the coexpression network approach identify novel pathway connections of molecular processes affected in Porto-sinusoidal vascular disease

PONE-D-25-56812R2

Dear Dr. Ehrhart,

We’re pleased to inform you that your manuscript has been judged scientifically suitable for publication and will be formally accepted for publication once it meets all outstanding technical requirements.

Kind regards,

Cheung Kenneth Chat Pan

Academic Editor

PLOS One
---

## [Editor Report · Acceptance letter]

PONE-D-25-56812R2

PLOS One

Dear Dr. Ehrhart,

I'm pleased to inform you that your manuscript has been deemed suitable for publication in PLOS One. Congratulations! Your manuscript is now being handed over to our production team.

Kind regards,

on behalf of

Dr. Cheung Kenneth Chat Pan

Academic Editor

PLOS One